# VEGF-C promotes the development of lymphatics in bone and bone loss

Devon Hominick[1†], Asitha Silva[1†], Noor Khurana[1], Ying Liu[2], Paul C Dechow[2], Jian Q Feng[2], Bronislaw Pytowski[3], Joseph M Rutkowski[4], Kari Alitalo[5], Michael T Dellinger[1,6,7]*

[1]Hamon Center for Therapeutic Oncology Research, UT Southwestern Medical Center, Dallas, United States; [2]Biomedical Sciences, Texas A&M College of Dentistry, Dallas, United States; [3]Eli Lilly and Company, New York, United States; [4]Division of Lymphatic Biology, Department of Medical Physiology, Texas A&M College of Medicine, Texas, United States; [5]Wihuri Research Institute and Translational Cancer Biology Program, Biomedicum Helsinki, University of Helsinki, Helsinki, Finland; [6]Hamon Center for Regenerative Science and Medicine, UT Southwestern Medical Center, Dallas, United States; [7]Division of Surgical Oncology, Department of Surgery, UT Southwestern Medical Center, Dallas, United States

*For correspondence:
michael.dellinger@
utsouthwestern.edu

[†]These authors contributed equally to this work

**Abstract** Patients with Gorham-Stout disease (GSD) have lymphatic vessels in their bones and their bones gradually disappear. Here, we report that mice that overexpress VEGF-C in bone exhibit a phenotype that resembles GSD. To drive VEGF-C expression in bone, we generated *Osx-tTA;TetO-Vegfc* double-transgenic mice. In contrast to *Osx-tTA* mice, *Osx-tTA;TetO-Vegfc* mice developed lymphatics in their bones. We found that inhibition of VEGFR3, but not VEGFR2, prevented the formation of bone lymphatics in *Osx-tTA;TetO-Vegfc* mice. Radiological and histological analysis revealed that bones from *Osx-tTA;TetO-Vegfc* mice were more porous and had more osteoclasts than bones from *Osx-tTA* mice. Importantly, we found that bone loss in *Osx-tTA; TetO-Vegfc* mice could be attenuated by an osteoclast inhibitor. We also discovered that the mutant phenotype of *Osx-tTA;TetO-Vegfc* mice could be reversed by inhibiting the expression of VEGF-C. Taken together, our results indicate that expression of VEGF-C in bone is sufficient to induce the pathologic hallmarks of GSD in mice.

DOI: https://doi.org/10.7554/eLife.34323.001

## Introduction

Lymphatic vessels perform several essential functions in the body. Lymphatic vessels absorb intestinal lipids, transport immune cells, and return fluid and macromolecules to the blood vasculature (*Witte et al., 1997*; *Zheng et al., 2014*). Most soft tissues in the body have lymphatic vessels (*Witte et al., 1997*). In contrast, hard tissues such as enamel, dentin, and bone do not have lymphatic vessels. Gorham-Stout disease (GSD) is a sporadic disease that is characterized by the presence of lymphatics in bone and bone loss (*Gorham and Stout, 1955*; *Dellinger et al., 2014*). J.B.S. Jackson is credited with publishing the first case of GSD, which appeared in the Boston Medical and Surgical Journal in 1838 (*Jackson, 1838*). In his paper entitled 'A Boneless Arm', Jackson described a patient whose humerus had completely disappeared (*Jackson, 1838*). Over 100 years later, Gorham and colleagues published a paper that described two additional cases of massive bone loss (*Gorham et al., 1954*). They were able to analyze multiple affected bones from one patient and found that the bones were filled with irregular vessels (*Gorham et al., 1954*). This interesting observation inspired Gorham and Stout to review additional cases of massive bone loss. Over several

years, Gorham and Stout were able to acquire biopsy specimens from seven patients seen by other physicians (*Gorham and Stout, 1955*). They observed abnormal vessels in each of the bone biopsy specimens they analyzed (*Gorham and Stout, 1955*). This finding led Gorham and Stout to report that massive bone loss is associated with the growth of vessels in bone (*Gorham and Stout, 1955*). It is now known that bones from GSD patients are filled with lymphatic vessels (*Lala et al., 2013*; *Ozeki et al., 2016*; *Edwards et al., 2008*).

Since Gorham and Stout's landmark publication, more than 300 cases of GSD have been described (*Dellinger et al., 2014*). These case reports have further defined the clinical characteristics of the disease. GSD affects males and females equally and is most commonly diagnosed in children and young adults (*Dellinger et al., 2014*). The course of this disease is often unpredictable. In some patients, the disease progresses slowly over a period of years, whereas in others, it progresses rapidly over a period of months (*Dellinger et al., 2014*). In severe cases, the disease progresses until entire bones are lost and replaced by fibrous tissue. The disease is reported to arrest and stabilize after a variable period of activity. Importantly, affected bones do not regenerate after the disease has arrested (*Dellinger et al., 2014*). Although GSD can affect any bone in the body, it most frequently affects the ribs and vertebrae (*Dellinger et al., 2014*; *Lala et al., 2013*). Unfortunately, thoracic involvement is associated with a poor prognosis because these patients tend to develop chylothorax, a complication that can cause respiratory distress, failure, and death (*Lala et al., 2013*; *Ludwig et al., 2016*). Despite advances in the understanding of the clinical features of the disease, the cause of GSD remains unknown and there are no standard treatments for this disease.

Vascular endothelial growth factor-C (VEGF-C) is the principle driver of lymphangiogenesis. Full-length VEGF-C undergoes proteolytic processing, which increases its affinity for VEGFR2 and VEGFR3 (*Joukov et al., 1996*; *Joukov et al., 1997*). Studies with genetically engineered mice have revealed that loss of VEGF-C impairs the development of the lymphatic vasculature. Mice that completely lack VEGF-C do not form lymphatics, whereas mice that lack a single copy of the VEGF-C gene develop fewer lymphatics than wildtype mice (*Dellinger et al., 2007*; *Karkkainen et al., 2004*). Conversely, tissue-specific overexpression of VEGF-C has been shown to induce lymphangiogenesis in the pancreas (*Mandriota et al., 2001*), lungs (*Yao et al., 2014*), and skin (*Jeltsch et al., 1997*; *Lohela et al., 2008*). In the present study, we characterize the effect of VEGF-C overexpression in bone on lymphatics and bone structure.

## Results

### VEGF-C induces the formation of bone lymphatics

Osterix (**Osx**) is a transcription factor expressed by chondrocytes, osteocytes, and osteoblasts (*Zhou et al., 2010*). A transgenic mouse line was previously created in which the expression of a tetracycline transactivator (tTA) cassette was placed under the control of the Osx promoter (*Rodda and McMahon, 2006*). This transgenic mouse line also has a *TetO-Cre::GFP* cassette located immediately downstream of the tTA cassette (*Rodda and McMahon, 2006*). To identify cell types that display tTA activity in *Osx-tTA-TetO-Cre::GFP* mice, we characterized the expression pattern of the Cre::GFP fusion protein with an anti-green fluorescent protein (GFP) antibody. GFP was expressed by chondrocytes, osteocytes, and osteoblasts (*Figure 1—figure supplement 1*). GFP was not expressed by cells in the kidney, liver, lung, pancreas, skeletal muscle, or spleen (*Figure 1—figure supplement 1*). These data show that tTA activity in *Osx-tTA-TetO-Cre::GFP* mice (herein referred to as *Osx-tTA*) is restricted to bone.

To induce VEGF-C expression in bone, we bred *Osx-tTA* transgenic mice with *TetO-Vegfc* transgenic mice (*Figure 1—figure supplement 2*). No viable *Osx-tTA;TetO-Vegfc* offspring were obtained from these crosses. To determine whether *Osx-tTA;TetO-Vegfc* mice died during embryonic development, we analyzed mice on embryonic day (E)12.5, E14.5, and E16.5. *Osx-tTA;TetO-Vegfc* mice were grossly indistinguishable from control littermates on E12.5. However, *Osx-tTA; TetO-Vegfc* mice were severely edematous and displayed enlarged lymphatics on E14.5. All *Osx-tTA;TetO-Vegfc* mice were in the process of being resorbed on E16.5 (*Figure 1—figure supplement 2*). These data indicate that VEGF-C expression by bone cells during embryonic development has a lethal effect. To overcome this lethal effect, we bred *Osx-tTA* mice with *TetO-Vegfc* mice and placed mice on doxycycline water from E0.5 to E18.5. Mice were then placed on normal water to induce

the expression of VEGF-C during postnatal development. To determine whether VEGF-C expression by bone cells induced the formation of bone lymphatics, we stained femurs from 21, 28, and 35-day-old *Osx-tTA* and *Osx-tTA;TetO-Vegfc* mice with an anti-podoplanin antibody. Lymphatic vessels were not present in femurs from 21, 28, or 35-day-old *Osx-tTA* mice. Similarly, lymphatic vessels were not present in femurs from 21-day-old *Osx-tTA;TetO-Vegfc* mice. In contrast, lymphatic vessels were present in femurs from 28-day-old *Ox-tTA;TetO-Vegfc* mice. However, lymphatic vessels were restricted to cortical bone (the dense outer shell of bone). In 35-day-old *Osx-tTA;TetO-Vegfc* mice, lymphatic vessels were located in cortical bone, trabecular bone, and the marrow cavity (*Figure 1*). Tibias and ribs from 35-day-old *Osx-tTA;TetO-Vegfc* mice also had lymphatics (*Figure 1—figure supplement 3*). These vessels expressed Lyve-1, another marker of lymphatic endothelium (*Figure 1—figure supplement 3*). Although 35-day-old *Osx-tTA;TetO-Vegfc* mice had hyperplastic lymphatics in bones, lymphatics in soft tissues (kidney, liver, lungs, and pancreas) appeared normal and the circulating level of VEGF-C was not elevated (*Figure 1—figure supplement 4*). These data indicate that expression of VEGF-C in bone induces the formation of bone lymphatics, but not the growth of lymphatics in distant tissues.

## VEGFR3 signaling is required for the formation of bone lymphatics

VEGF-C activates VEGFR2 and VEGFR3 (*Joukov et al., 1996*). However, VEGF-C is thought to primarily induce lymphangiogenesis by activating VEGFR3 (*Veikkola et al., 2001*). To determine whether VEGF-C activation of VEGFR2 or VEGFR3 was required for the formation of bone lymphatics, we treated 21-day-old *Osx-tTA;TetO-Vegfc* mice with vehicle, DC101 (VEGFR2 function-blocking antibody), or mF4-31C1 (VEGFR3 function-blocking antibody) for 2 weeks. The density of bone lymphatics was modestly reduced in DC101-treated mice. In contrast, bone lymphatics failed to form in mF4-31C1-treated mice (*Figure 2*). These data show that VEGFR3 signaling is required for the development of bone lymphatics in *Osx-tTA;TetO-Vegfc* mice.

In contrast to murine VEGF-C, murine VEGF-D only activates VEGFR3 (*Baldwin et al., 2001*). A transgenic mouse strain was recently created in which the expression of murine VEGF-D is controlled by doxycycline (*Lammoglia et al., 2016*). To determine whether murine VEGF-D could induce the formation of bone lymphatics, we generated *Osx-tTA;TetO-Vegfd* mice. *Osx-tTA;TetO-Vegfd* mice received doxycycline water from E0.5 to E18.5 and then normal water from E18.5 to postnatal day (P) 35. Importantly, femurs from *Osx-tTA;TetO-Vegfd* mice were filled with lymphatic vessels (*Figure 2*). These data show that activation of VEGFR3 is sufficient for the formation of bone lymphatics.

## Bone loss in *Osx-tTA;TetO-Vegfc* mice is mediated by osteoclasts

To characterize the effect of VEGF-C expression by bone cells on bone structure, we X-rayed femurs from 21, 28, and 35-day-old *Osx-tTA* and *Osx-tTA;TetO-Vegfc* mice. Femurs from 21-day-old *Osx-tTA;TetO-Vegfc* mice appeared normal and were indistinguishable from femurs from *Osx-tTA* mice (*Figure 3—figure supplement 1*). In contrast, femurs from 28- and 35-day-old *Osx-tTA;TetO-Vegfc* mice had a moth-eaten appearance (*Figure 3—figure supplement 1*). The development of this phenotype coincided with the formation of lymphatics in cortical bone. To further characterize the structure of cortical bone, we took µCT images of femurs and ribs from 35-day-old *Osx-tTA* and *Osx-tTA;TetO-Vegfc* mice. This revealed that bones from *Osx-tTA;TetO-Vegfc* mice were significantly more porous than bones from *Osx-tTA* mice (*Figure 3*). To determine whether the structural defects in *Osx-tTA;TetO-Vegfc* bones affected bone strength, we performed a three-point bending assay. The three-point bending assay revealed that femurs from *Osx-tTA;TetO-Vegfc* mice were significantly weaker than femurs from *Osx-tTA* mice (*Figure 3*).

Next, we set out to delineate the cause of bone loss in *Osx-tTA;TetO-Vegfc* mice. To determine whether lymphatic endothelial cells (LECs) could directly degrade bone, we cultured RAW264.7 cells (osteoclast precursor cell line) and primary human LECs in separate wells of an osteo-assay surface plate. The osteo-assay surface plate has a calcium-phosphate coating that mimics bone. After culturing cells for 72 hr, we observed numerous resorption pits in wells containing differentiated RAW264.7 cells (osteoclasts), but not in wells containing LECs (*Figure 3—figure supplement 2*). This finding suggests that LECs do not directly degrade bone.

Osteoclasts are multi-nucleated cells that degrade bone and display tartrate-resistant acid phosphatase (TRAP) activity. To determine whether there was a difference in the number of osteoclasts

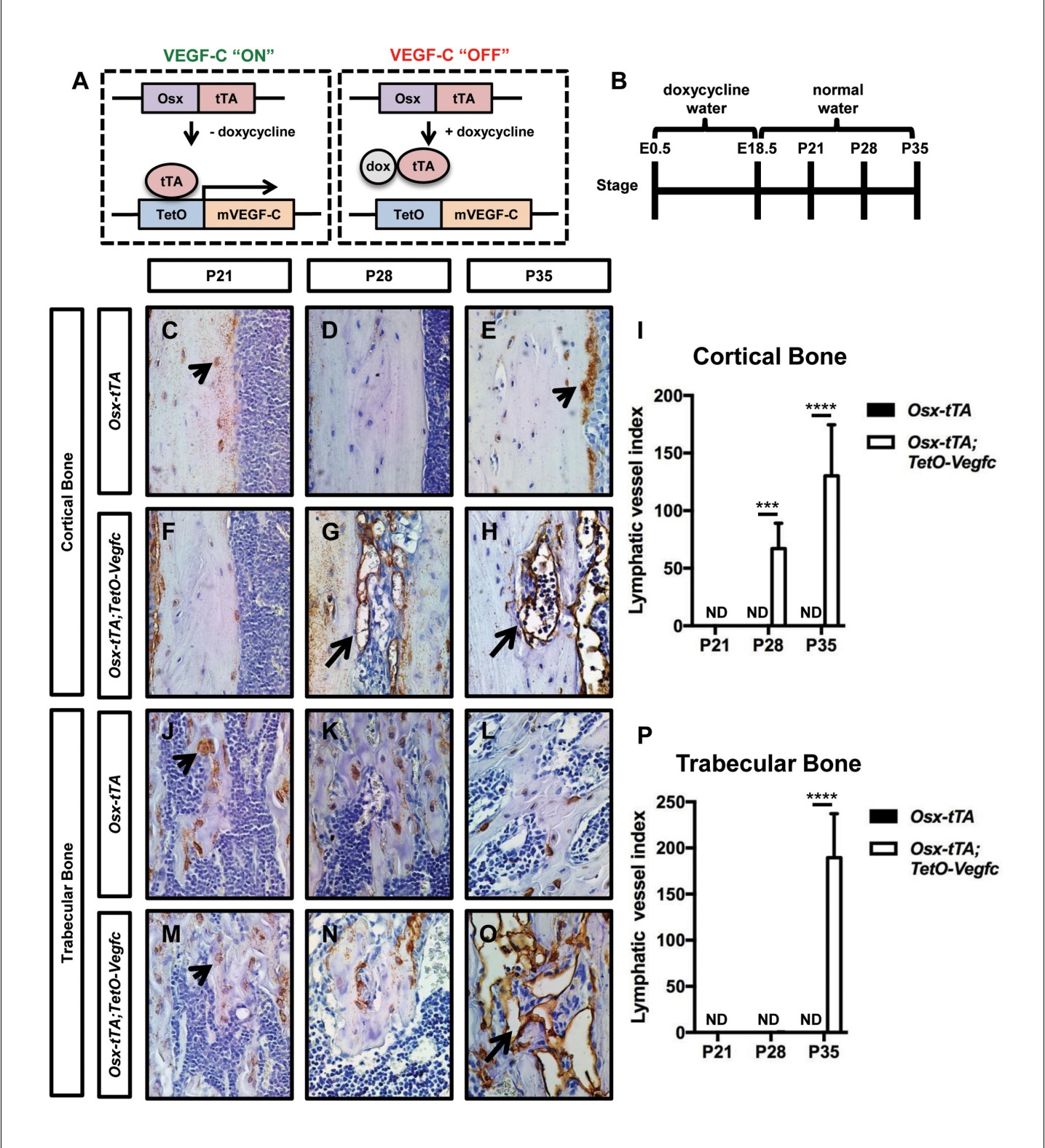

**Figure 1.** *Osx-tTA;TetO-Vegfc* mice develop lymphatics in their bones. (**A**) Schematic of the Tet-Off system used to express VEGF-C in bone. Doxycycline inhibits the expression of VEGF-C. (**B**) Schematic showing when mice received normal water and doxycycline water. *Osx-tTA* and *Osx-tTA; TetO-Vegfc* mice received doxycycline water from E0.5 to E18.5 and then normal water from E18.5 to P35. (**C–H**) Representative images of cortical bone in *Osx-tTA* and *Osx-tTA;TetO-Vegfc* femurs. Sections were stained with an anti-podoplanin antibody. Arrowheads point to podoplanin-positive osteocytes. Arrows point to podoplanin-positive lymphatics. (**I**) Graph showing lymphatic vessel index values for cortical bone in P21 (0 ± 0.0; n = 3), P28

*Figure 1 continued on next page*

*Figure 1 continued*

(0 ± 0.0; n = 5), and P35 (0 ± 0.0; n = 6) *Osx-tTA* mice and in P21 (0 ± 0.0; n = 4), P28 (67 ± 22.06; n = 4), and P35 (130.3 ± 44.35; n = 4) *Osx-tTA;TetO-Vegfc* mice. (J–O) Representative images of trabecular bone in *Osx-tTA* and *Osx-tTA;TetO-Vegfc* femurs. Sections were stained with an anti-podoplanin antibody. Arrowheads point to podoplanin-positive osteocytes. Arrow points to podoplanin-positive lymphatics. (P) Graph showing lymphatic vessel index values for trabecular bone in P21 (0 ± 0.0; n = 3), P28 (0 ± 0.0; n = 5), and P35 (0 ± 0.0; n = 6) *Osx-tTA* mice and P21 (0 ± 0.0; n = 4), P28 (0.167 ± 0.4082; n = 6), and P35 (189.5 ± 47.7; n = 4) *Osx-tTA;TetO-Vegfc* mice. (***p<0.001, ****p<0.0001, unpaired student's T-test). ND = Not Detected.

DOI: https://doi.org/10.7554/eLife.34323.002

The following figure supplements are available for figure 1:

**Figure supplement 1.** Expression pattern of GFP in *Osx-tTA-TetO-tTA-Cre::GFP* mice.
DOI: https://doi.org/10.7554/eLife.34323.003
**Figure supplement 2.** Expression of VEGF-C during embryonic development causes embryonic lethality.
DOI: https://doi.org/10.7554/eLife.34323.004
**Figure supplement 3.** Bone lymphatics express Lyve-1.
DOI: https://doi.org/10.7554/eLife.34323.005
**Figure supplement 4.** Lymphatics in soft tissues in *Osx-tTA;TetO-Vegfc* mice appear normal.
DOI: https://doi.org/10.7554/eLife.34323.006

between *Osx-tTA* and *Osx-tTA;TetO-Vegfc* mice, we stained bones from 35-day-old mice for TRAP activity. This staining revealed that there were significantly more osteoclasts on the surface of cortical bone in *Osx-tTA;TetO-Vegfc* mice than in *Osx-tTA* mice (*Figure 3*). To extend this finding, we also measured the circulating levels of C-terminal telopeptide of type I collagen (CTX-1) in *Osx-tTA* and *Osx-tTA;TetO-Vegfc* mice. CTX-1 is a small peptide generated by osteoclast-mediated cleavage of collagen I (*Garnero et al., 2003*). Importantly, CTX-1 was significantly higher in *Osx-tTA;TetO-Vegfc* mice than *Osx-tTA* mice (*Figure 3*). Together, these data show that the number and activity of osteoclasts are increased in *Osx-tTA;TetO-Vegfc* mice.

Zoledronic acid (also known as Zometa) is a drug that accumulates in bone and prevents bone resorption by inhibiting farnesyl pyrophosphate synthase activity in osteoclasts (*Clézardin, 2013*). To determine whether osteoclasts promote bone resorption in *Osx-tTA;TetO-Vegfc* mice, we treated 21-day-old mice with either vehicle or zoledronic acid for 2 weeks. The density of bone lymphatics was not significantly different between zoledronic acid-treated and vehicle-treated *Osx-tTA;TetO-Vegfc* mice (*Figure 4*). However, the number of osteoclasts and porosity of femurs were significantly lower in zoledronic acid-treated mice than vehicle-treated mice (*Figure 4*). These data indicate that bone loss in *Osx-tTA;TetO-Vegfc* mice is driven by osteoclasts.

To determine whether inhibition of lymphangiogenesis could prevent osteoclastogenesis and bone loss in *Osx-tTA;TetO-Vegfc* mice, we analyzed femurs from vehicle and mF4-31C1-treated mice. We focused our analysis on mF4-31C1-treated mice because these mice do not develop bone lymphatics (*Figure 2*). We found that femurs from mF4-31C1-treated mice had significantly fewer osteoclasts than femurs from vehicle-treated mice (*Figure 5*). Additionally, we found that femurs from mF4-31C1-treated mice were significantly less porous than femurs from vehicle-treated mice (*Figure 5*). These data show that inhibition of lymphangiogenesis can prevent osteoclastogenesis and bone loss in *Osx-tTA;TetO-Vegfc* mice.

## Inhibition of VEGF-C expression reverses the mutant phenotype of *Osx-tTA;TetO-Vegfc* mice

Next, we set out to determine whether the mutant phenotype of *Osx-tTA;TetO-Vegfc* mice was reversible. We first characterized the reversibility of VEGF-C expression in *Osx-tTA;TetO-Vegfc* mice. We collected RNA from tibias from *Osx-tTA* mice, *Osx-tTA;TetO-Vegfc* mice that received normal water from E18.5 to P35, and *Osx-tTA;TetO-Vegfc* mice that received normal water from E18.5 to P35 and then doxycycline water for either 3 or 7 days (*Figure 6*). The expression of VEGF-C mRNA was then evaluated by qPCR. VEGF-C mRNA levels were approximately 350-fold higher in *Osx-tTA;TetO-Vegfc* mice than *Osx-tTA* mice. Importantly, VEGF-C mRNA levels in *Osx-tTA;TetO-Vegfc* mice returned to normal within 3 days of being placed back on doxycycline water (*Figure 6*). These data indicate that doxycycline rapidly inhibits the expression of VEGF-C in *Osx-tTA;TetO-Vegfc* mice.

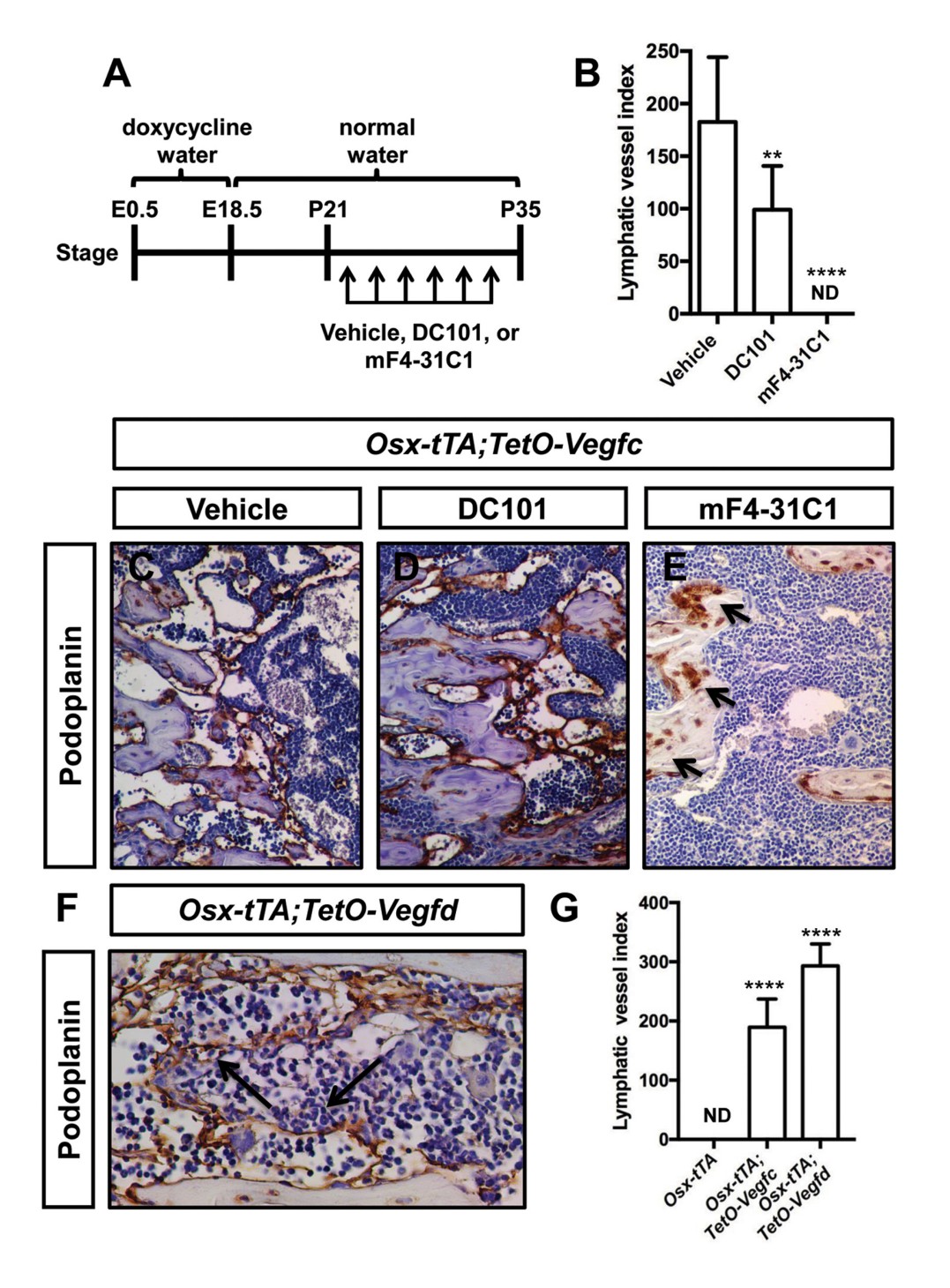

**Figure 2.** VEGFR3 signaling promotes the formation of bone lymphatics. (A) Schematic showing when mice received normal water and doxycycline water. *Osx-tTA;TetO-Vegfc* mice were treated (3x/week) with vehicle, DC101 (VEGFR2 function-blocking antibody), or mF4-31C1 (VEGFR3 function-blocking antibody) from P21 to P35. (B) Graph showing lymphatic index values for vehicle-treated (182.6 ± 61.56; n = 7), DC101-treated (99 ± 41.68; n = 7), and mF4-31C1-treated mice (0 ± 0.0; n = 5). (**p<0.01, ****p<0.0001, ANOVA followed by Dunnett's multiple comparisons test. Values were tested against values for vehicle-treated mice). (C–E) Representative images of femurs stained with an anti-podoplanin antibody. The femurs are from P35 mice. Arrows in panel E point to podoplanin-positive osteocytes. (F) Representative image of a femur from an *Osx-tTA;TetO-Vegfd* mouse that was stained with an anti-podoplanin antibody. Arrows point to podoplanin-positive lymphatics in the bone marrow. (G) Graph showing lymphatic vessel index values for trabecular bone in P35 *Osx-tTA* mice (0 ± 0; n = 6), *Osx-tTA;TetO-Vegfc* mice (189.5 ± 47.7; n = 4), and *Osx-tTA;TetO-Vegfd* mice

*Figure 2 continued on next page*

*Figure 2 continued*

(293 ± 37.24; n = 3). (****p<0.0001, ANOVA followed by Dunnett's multiple comparisons test. Values were tested against values for *Osx-tTA* mice).
ND = Not Detected.
DOI: https://doi.org/10.7554/eLife.34323.007

To assess the reversibility of the lymphatic phenotype of *Osx-tTA;TetO-Vegfc* mice, we analyzed ribs from *Osx-tTA* mice, *Osx-tTA;TetO-Vegfc* mice that received normal water from E18.5 to P35, and *Osx-tTA;TetO-Vegfc* mice that received normal water from E18.5 to P35 and then doxycycline water for either 3, 7, 28, or 56 days (*Figure 6*). *Osx-tTA* mice did not have lymphatic vessels in their ribs and possessed a sparse network of lymphatics in their periosseous muscle (*Figure 6*). In contrast, *Osx-tTA;TetO-Vegfc* mice that received normal water from E18.5 to P35 had numerous lymphatic vessels in their ribs and an expanded network of lymphatics in their periosseous muscle (*Figure 6*). To our surprise, the abnormal lymphatic vessels in the ribs, but not in the periosseous muscle, gradually disappeared in *Osx-tTA;TetO-Vegfc* mice that received doxycycline (*Figure 6*). These data show that irregular lymphatic vessels in bone, but not in periosseous muscle, depend on continued VEGF-C signaling for their existence.

We next set out to characterize the reversibility of the bone phenotype of *Osx-tTA;TetO-Vegfc* mice. We X-rayed ribs from *Osx-tTA* mice, *Osx-tTA;TetO-Vegfc* mice that received normal water from E18.5 to P35, and *Osx-tTA;TetO-Vegfc* mice that received normal water from E18.5 to P35 and then doxycycline water for 28 days. Similar to what we observed for femurs (*Figure 3—figure supplement 1*) ribs in *Osx-tTA;TetO-Vegfc* mice had a moth-eaten appearance (*Figure 7*). Amazingly, inhibition of VEGF-C expression caused ribs in *Osx-tTA;TetO-Vegfc* mice to switch from a moth-eaten appearance to a normal appearance (*Figure 7*). To further explore the effect of VEGF-C inhibition on bone structure in *Osx-tTA;TetO-Vegfc* mice, we evaluated cortical bone porosity in femur sections. Doxycycline caused cortical bone porosity in *Osx-tTA;TetO-Vegfc* mice to revert back to normal. We also found that the number of osteoclasts and the circulating levels of CTX-1 in *Osx-tTA; TetO-Vegfc* mice returned to normal after exposure to doxycycline (*Figure 7*). Together, these data show that the bone phenotype of *Osx-tTA;TetO-Vegfc* mice is reversible.

### *Osx-tTA;TetO-Vegfc* mice develop chylothorax

Chylothorax is frequently observed in GSD patients with thoracic involvement and is the leading cause of mortality of GSD patients (*Ludwig et al., 2016*). During the course of our experiments we discovered that *Osx-tTA;TetO-Vegfc* mice die prematurely (*Figure 8*). When we examined moribund *Osx-tTA;TetO-Vegfc* mice, we found that their chest cavity was filled with fluid. The color of this fluid ranged from milky white to light pink. The triglyceride content of the fluid was greater than 110 mg/dl, which indicated that the effusion fluid was chyle (*Figure 8*). Chyle is lymph rich in fat that flows through the thoracic duct. To identify the route by which chyle escaped the thoracic duct in *Osx-tTA;TetO-Vegfc* mice, we injected the mesenteric lymph node of 35-day-old mice with Evans blue dye (EBD). EBD was confined to the thoracic duct in *Osx-tTA* mice. In contrast, EBD spilled from the thoracic duct into lymphatics in the periosseous muscle in *Osx-tTA;TetO-Vegfc* mice. EBD appeared to leak from the irregular lymphatics along the chest wall. These findings indicate that the lymphatics that form in the periosseous muscle do not function properly.

## Discussion

Patients with GSD have lymphatics in their bones and their bones gradually disappear. In this study, we show that overexpression of VEGF-C by bone cells induces the formation of bone lymphatics and bone loss. Taken together, our findings demonstrate that expression of VEGF-C in bone is sufficient to induce the pathologic hallmarks of GSD in mice.

VEGF-C has been shown to promote lymphangiogenesis in several different pathological settings. However, investigators have only recently started to examine VEGF-C levels in patients with GSD. The circulating level of VEGF-C was found to be slightly elevated in 1 GSD patient (*Brodszki et al., 2011*), but not in two other GSD patients (*Brodszki et al., 2011*; *Baud et al., 2015*). We found that the local level, but not the circulating level, of VEGF-C was elevated in *Osx-tTA;TetO-Vegfc* mice. In

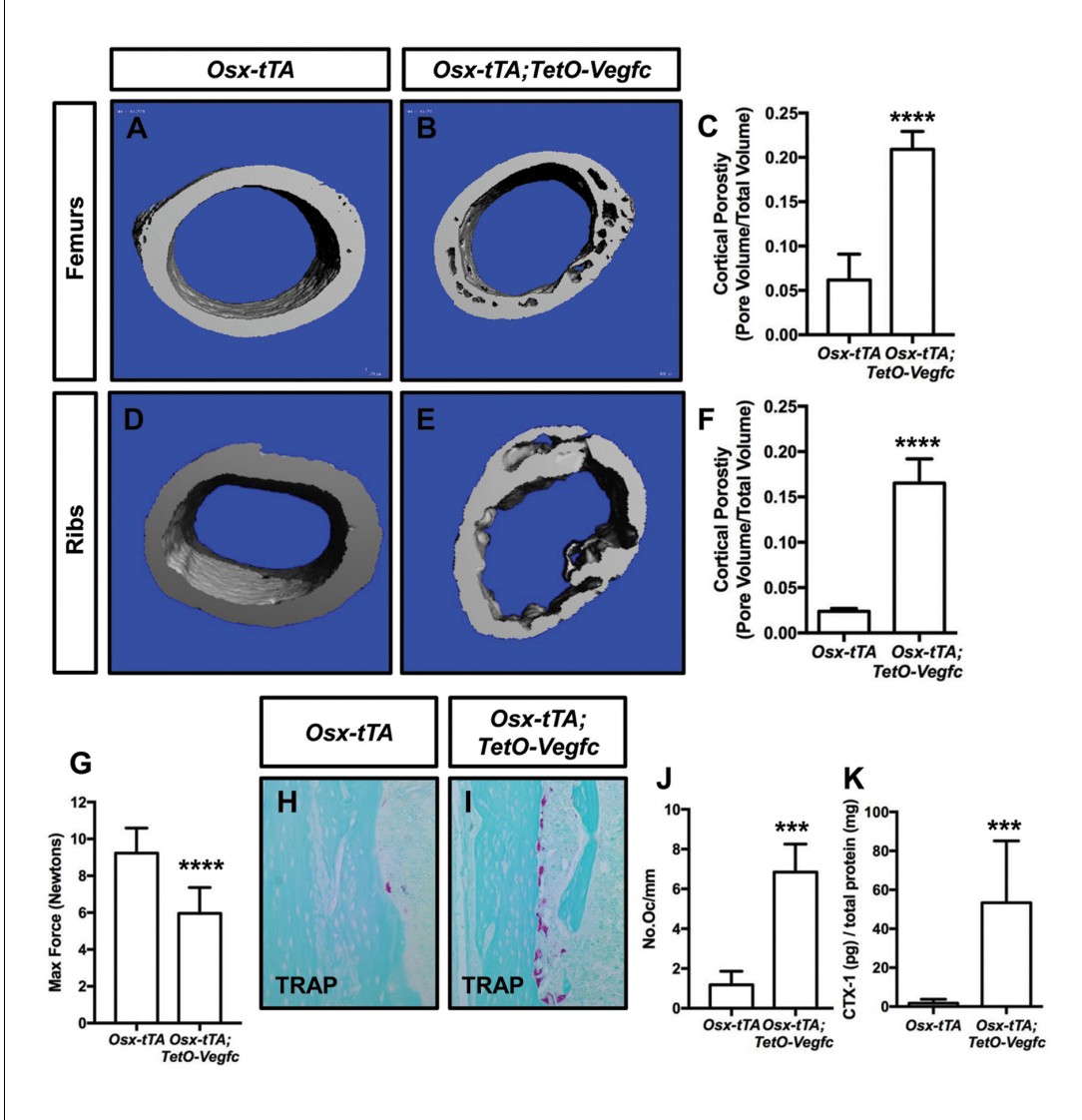

**Figure 3.** *Osx-tTA;TetO-Vegfc* mice have more porous bones and osteoclasts than *Osx-tTA* mice. (A,B) Representative µCT images of femurs from *Osx-tTA* and *Osx-tTA;TetO-Vegfc* mice. (C) Graph showing cortical bone porosity for femurs from *Osx-TA* ($0.062 \pm 0.0292$, n = 5) and *Osx-tTA;TetO-Vegfc* ($0.209 \pm 0.0204$, n = 6) mice. (D,E) Representative µCT images of ribs from *Osx-tTA* and *Osx-tTA;TetO-Vegfc* mice. (F) Graph showing cortical bone porosity for ribs from *Osx-TA* ($0.024 \pm 0.00293$, n = 4) and *Osx-tTA;TetO-Vegfc* ($0.165 \pm 0.0265$, n = 4) mice. (G) Graph showing results from the three-point bending assay. Less force was required to break bones from *Osx-TA;TetO-Vegfc* mice ($5.96 \pm 1.404$, n = 11) than *Osx-tTA* mice ($9.231 \pm 1.355$, n = 9) mice. (H,I) Representative images of TRAP-stained femurs from *Osx-tTA* and *Osx-tTA;TetO-Vegfc* mice. (J) Graph showing the number of osteoclasts per mm of bone for *Osx-tTA* ($1.18 \pm 0.6818$; n = 4) and *Osx-tTA;TetO-Vegfc* ($6.84 \pm 1.413$; n = 4) mice. (K) Graph showing CTX-1 values for *Osx-tTA* ($1.7 \pm 2.045$; n = 8) and *Osx-tTA;TetO-Vegfc* ($53.3 \pm 31.8$; n = 7) mice. (***$p<0.001$, ****$p<0.0001$, unpaired student's T-test).

DOI: https://doi.org/10.7554/eLife.34323.008

The following figure supplements are available for figure 3:

**Figure supplement 1.** Bones from *Osx-tTA;TetO-Vegfc* mice have a moth-eaten appearance.

DOI: https://doi.org/10.7554/eLife.34323.009

**Figure supplement 2.** LECs do not degrade a calcium-phosphate matrix.

DOI: https://doi.org/10.7554/eLife.34323.010

the future, patient samples could be used to determine whether VEGF-C is locally elevated in affected tissues in GSD patients.

VEGF-C is a growth factor that promotes lymphangiogenesis in embryos and in adult tissues. VEGF-C activates VEGFR2 and VEGFR3 (*Joukov et al., 1996*). However, VEGF-C has a greater

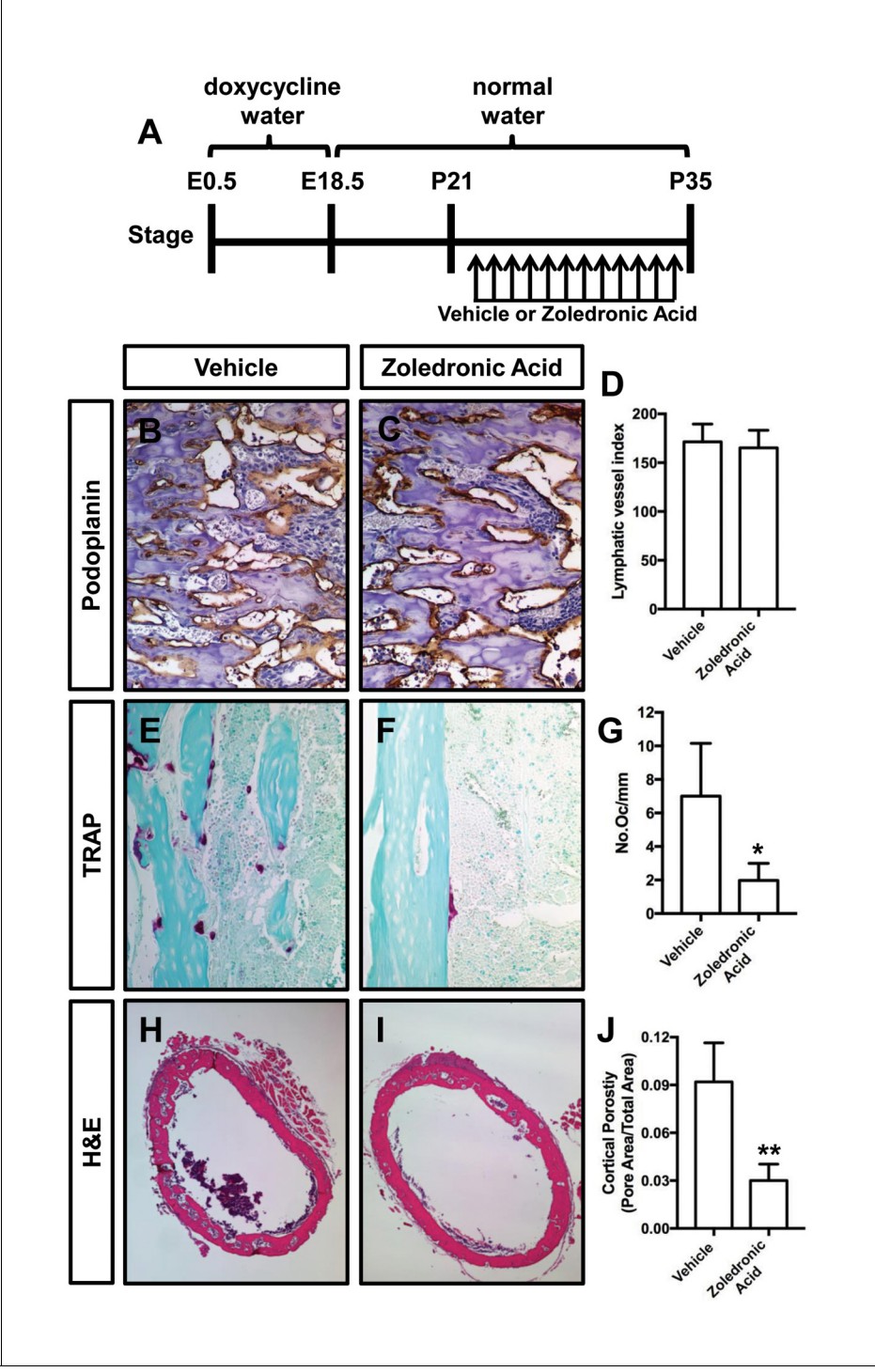

**Figure 4.** Zoledronic acid attenuates bone loss in *Osx-tTA;TetO-Vegfc* mice. (A) Schematic showing when mice received normal water and doxycycline water. *Osx-tTA;TetO-Vegfc* mice were treated (q.a.d.) with vehicle or zoledronic acid from P21 to P35. (B,C) Representative images of femurs stained with an anti-podoplanin antibody. (D) Graph showing lymphatic vessel index values for vehicle-treated (171.5 ± 18.18, n = 3) and zoledronic acid-treated (165.2 ± 18.04, n = 5) mice. (E,F) Representative images of TRAP stained femurs from vehicle-treated and zoledronic acid-treated mice. (G) Graph showing the number of osteoclasts per mm of bone for vehicle-treated (7.0 ± 3.15, n = 3) and zoledronic acid-treated (1.9 ± 1.9, n = 5) mice. (H,I) Representative images of H and E stained femurs from vehicle-treated and zoledronic acid-treated mice. (J) Graph showing cortical bone porosity of femurs for vehicle-treated (0.092 ± 0.0245, n = 3) and zoledronic acid-treated (0.030 ± 0.0103, n = 5) mice. (*p<0.05, **p<0.01, unpaired student's T-test).

DOI: https://doi.org/10.7554/eLife.34323.011

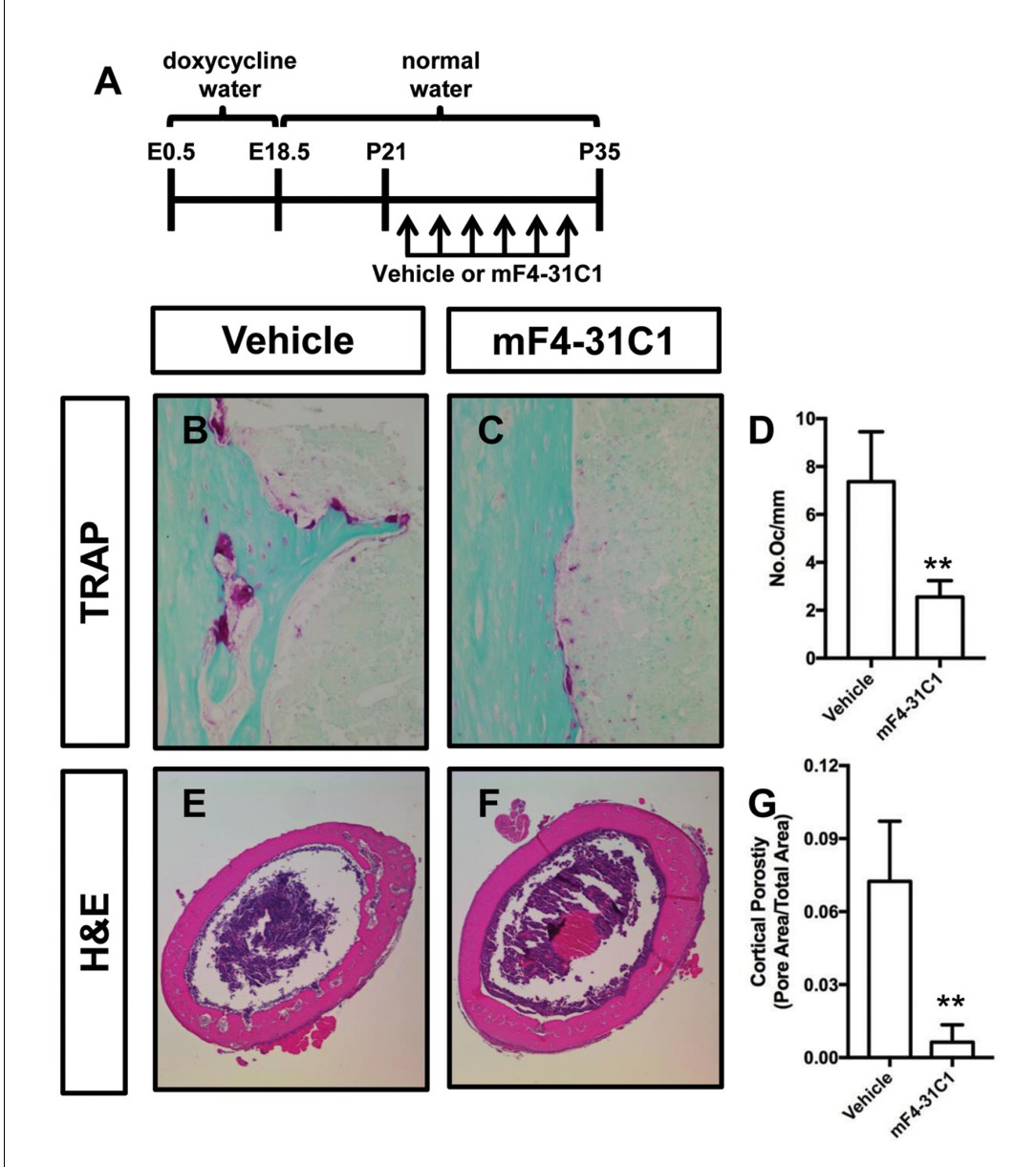

**Figure 5.** mF4-31C1 inhibits osteoclast formation and bone loss in *Osx-tTA;TetO-Vegfc* mice. (A) Schematic showing when mice received normal water and doxycycline water. *Osx-tTA;TetO-Vegfc* mice were treated (3x/week) with vehicle or mF4-31C1 (VEGFR3 function-blocking antibody) from P21 to P35. (B,C) Representative images of TRAP stained femurs from vehicle-treated and mF4-31C1-treated mice. (D) Graph showing the number of osteoclasts per mm of bone for vehicle-treated (7.37 ± 2.088, n = 5) and mF4-31C1-treated (2.552 ± 0.6893, n = 5) mice. (E,F) Representative images of H and E stained femurs from vehicle-treated and mF4-31C1-treated mice. (G) Graph showing cortical bone porosity of femurs for vehicle-treated (0.07244 ± 0.02468, n = 5) and mF4-31C1-treated (0.006375 ± 0.007087, n = 4) mice. (**p<0.01, unpaired student's T-test).

DOI: https://doi.org/10.7554/eLife.34323.012

affinity for VEGFR3 than VEGFR2 (*Joukov et al., 1997*). We found that mF4-31C1 (VEGFR3 inhibitor) was more effective at inhibiting lymphangiogenesis in *Osx-tTA;TetO-Vegfc* mice than DC101 (VEGFR2 inhibitor). Our findings are consistent with other studies that have reported that blockade or loss of VEGFR3 has a greater effect on lymphangiogenesis than blockade or loss of VEGFR2 (*Yao et al., 2014*; *Zarkada et al., 2015*; *Baluk et al., 2005*; *Yuen et al., 2011*).

LECs arise from multiple different sources. LECs come from pre-existing LECs (*Srinivasan et al., 2007*), blood endothelial cells (BECs) (*Srinivasan et al., 2007*; *Stanczuk et al., 2015*; *Martinez-*

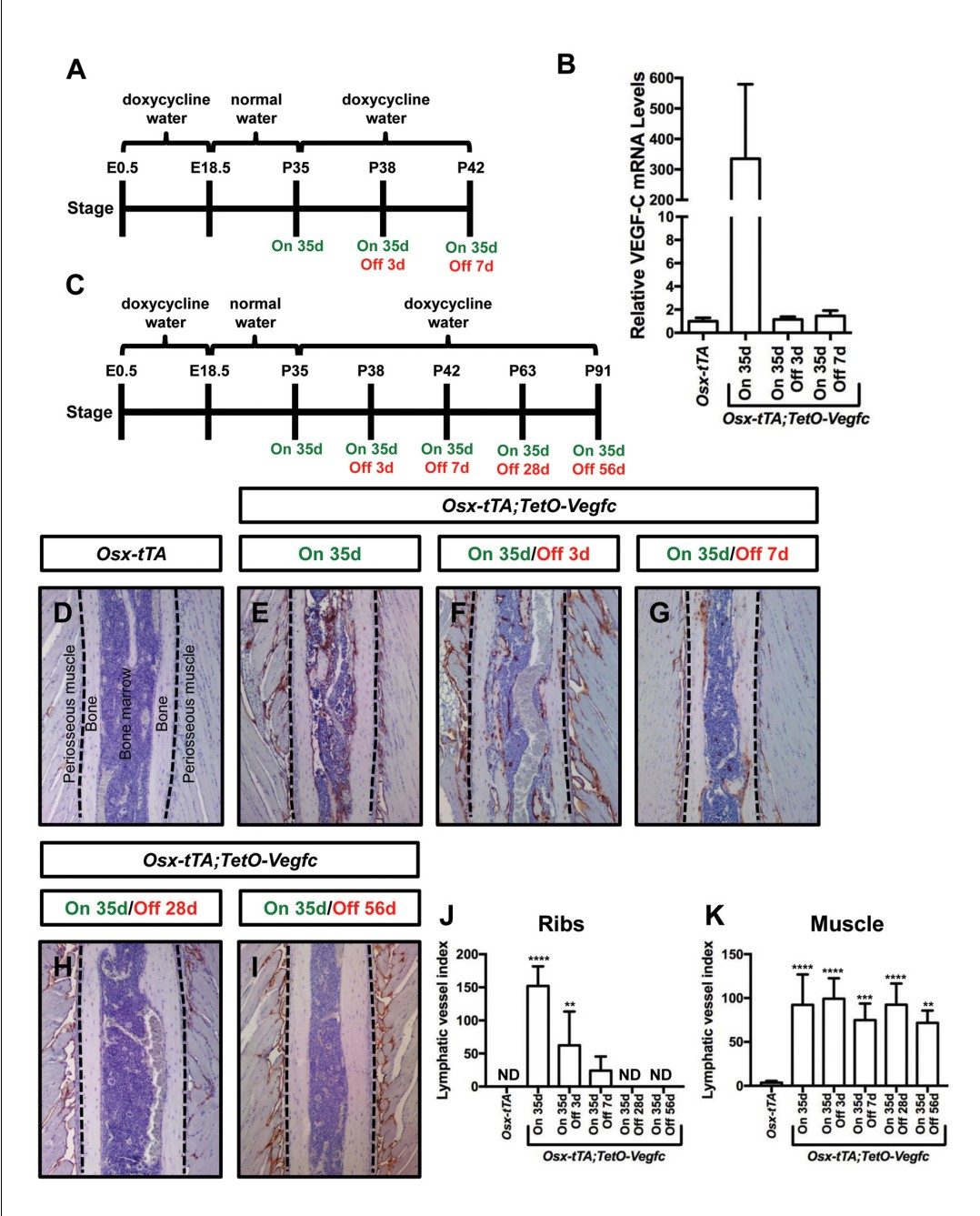

**Figure 6.** Bone lymphatics in *Osx-tTA;TetO-Vegfc* mice disappear following the withdrawal of VEGF-C. (A) Schematic showing when mice received normal water and doxycycline water. One cohort of *Osx-tTA;TetO-Vegfc* mice received normal water from E18.5 to P35 (On 35d). A second cohort of *Osx-tTA;TetO-Vegfc* mice received normal water from E18.5 to P35 and then doxycycline water from P35 to P38 (On 35d/Off 3d). A third cohort of *Osx-tTA;TetO-Vegfc* mice received normal water from E18.5 to P35 and then doxycycline water from P35 to P42 (On 35d/Off 7d). (B) Graph showing the relative VEGF-C mRNA levels in tibias from mice. (C) Schematic showing when mice received normal water and doxycycline water. *Osx-tTA;TetO-Vegfc* mice received normal water from E18.5 to P35 (On 35d) or normal water from E18.5 to P35 and then doxycycline water for 3 (On 35d/Off 3d), 7 (On 35d/ Off 7d), 28 (On 35d/Off 28d), or 56 days (On 35d/Off 56d). (D–I) Representative images of ribs stained with an anti-podoplanin antibody. The dashed lines separate the bone from the periosseous muscle. (J) Graph showing lymphatic vessel index values for ribs in *Osx-tTA* mice (0 ± 0.0; n = 5), *Osx-tTA; TetO-Vegfc* mice that received normal water for 35 days (152.5 ± 29.56; n = 5), and *Osx-tTA;TetO-Vegfc* mice that received normal water for 35 days and then doxycycline water for 3 (62.25 ± 51.7; n = 4), 7 (24.08 ± 21.26; n = 4), 28 (0 ± 0.0; n = 5) or 56 (0 ± 0.0; n = 3) days. (K) Graph showing lymphatic vessel index values for periosseous muscle in *Osx-tTA* mice (3.61 ± 1.974; n = 5), *Osx-tTA;TetO-Vegfc* mice that received normal water for 35 days (92.45 ± 34.63; n = 5), and *Osx-tTA;TetO-Vegfc* mice that received normal water for 35 days and then doxycycline water for 3 (99.29 ± 23.37; n = 4), 7

*Figure 6 continued on next page*

*Figure 6 continued*

(74.84 ± 18.98; n = 4), 28 (92.67 ± 24.2; n = 5) or 56 (72.17 ± 14.05; n = 3) days. (**p<0.01, ***p<0.001, ****p<0.0001, ANOVA followed by Dunnett's multiple comparisons test. Values were tested against values for *Osx-tTA* mice). ND = Not Detected.

DOI: https://doi.org/10.7554/eLife.34323.013

*Corral et al., 2015*), and from an unknown cellular source (*Martinez-Corral et al., 2015*). There is growing evidence that bone marrow-derived cells can also differentiate into LECs. Bone marrow transplantation experiments with GFP-positive donor mice suggest that bone-marrow-derived cells can incorporate into lymphatic vessels (*Religa et al., 2005*; *Jiang et al., 2008*). Additionally, mesenchymal stem cells isolated from bone are reported to express lymphatic markers such as Prox1 and Lyve-1 following exposure to VEGF-C (*Lee et al., 2010*). The origin of bone LECs is not known. Lineage-tracing studies with *Osx-tTA;TetO-Vegfc* mice could reveal whether LECs in bone arise from pre-existing LECs, BECs, or from a different cellular source.

We found that *Osx-tTA;TetO-Vegfc* mice had more osteoclasts than *Osx-tTA* mice. There are conflicting reports as to whether or not VEGF-C can directly stimulate osteoclast formation. VEGF-C was reported to promote osteoclast formation by one group (*Motokawa et al., 2013*) but not by another (*Zhang et al., 2008*). We found that VEGF-C did not induce or enhance osteoclast formation by RAW264.7 cells (data not shown). We also found that osteoclasts did not express VEGFR2 or VEGFR3 in vivo (data not shown). Therefore, other factors are likely responsible for stimulating osteoclast formation in *Osx-tTA;TetO-Vegfc* mice. Macrophage colony-stimulating factor (M-CSF) is a factor that promotes the development of osteoclasts (*Dai et al., 2002*). It was recently reported that LECs express a high level of M-CSF and that LEC-conditioned media can stimulate osteoclast formation (*Wang et al., 2017a2017*). Importantly, an M-CSF function-blocking antibody inhibited the ability of LEC-conditioned media to promote osteoclast formation (*Wang et al., 2017b*). Therefore, LECs in the bones of *Osx-tTA;TetO-Vegfc* mice may stimulate osteoclast development by secreting M-CSF.

The precise cause of bone loss in GSD is unclear. A prevailing view in the field is that osteoclasts promote excessive bone resorption in GSD. Osteoclasts have been detected in many different GSD patient samples (*Pazzaglia et al., 1997*; *Avelar et al., 2010*; *Bruder et al., 2009*; *Choma et al., 1987*; *Hammer et al., 2005*; *Hirayama et al., 2001*; *Lehmann et al., 2009*; *Möller et al., 1999*; *Silva, 2011*; *Spieth et al., 1997*; *Poirier, 1968*; *Jones et al., 1958*) and CTX-1 has been reported to be elevated in several patients (*Liu et al., 2016*). Zoledronic acid has been given to GSD patients in an attempt to prevent bone loss (*Avelar et al., 2010*; *Hagendoorn et al., 2006*; *Kuriyama et al., 2010*; *Leite et al., 2013*; *Ruggieri et al., 2011*; *Gem et al., 2014*; *Mignogna et al., 2005*; *Yerganyan et al., 2015*). Indeed, several case reports state that there is no further progression of the disease after the patient is treated with zoledronic acid (*Avelar et al., 2010*; *Kuriyama et al., 2010*; *Gem et al., 2014*; *Mignogna et al., 2005*; *Yerganyan et al., 2015*). However, it is difficult to assess the benefit of zoledronic acid in GSD patients because it is often given in combination with other treatments and it is usually unclear as to whether the osteolytic process was active or inactive at the start of treatment. We found that zoledronic acid attenuated bone loss in *Osx-tTA;TetO-Vegfc* mice. These data indicate that osteoclasts promote the destruction of bone in *Osx-tTA;TetO-Vegfc* mice and warrant further investigation into the effects of osteoclast inhibitors on GSD.

Several studies have shown that newly formed lymphatic vessels persist following the withdrawal of a growth-promoting stimulus. One study showed that abnormal lymphatic vessels persisted in the airways of *CCSP-rtTA;TetO-Vegfc* mice for up to 19 months after the withdrawal of VEGF-C (*Yao et al., 2014*). Another study showed that abnormal lymphatic vessels persisted in the skin of *K14-rtTA;TetO-Vegfc* mice for up to 6 months after the withdrawal of VEGF-C (*Lohela et al., 2008*). Additionally, it was reported that global inactivation of VEGF-C in adult mice only caused lymphatic vessels in the intestine to atrophy (*Nurmi et al., 2015*). Together, these findings suggest that lymphatic vessels in most tissues do not spontaneously regress following the withdrawal of VEGF-C. We found that newly formed lymphatics in bone, but not in periosseous muscle, disappeared following the withdrawal of VEGF-C. One possible explanation for this finding is that bone lacks lymphatic maintenance factors. An alternative explanation for this finding is that bone possesses lymphatic inhibitory factors whose effects can be overcome by overexpressing VEGF-C. Future experiments

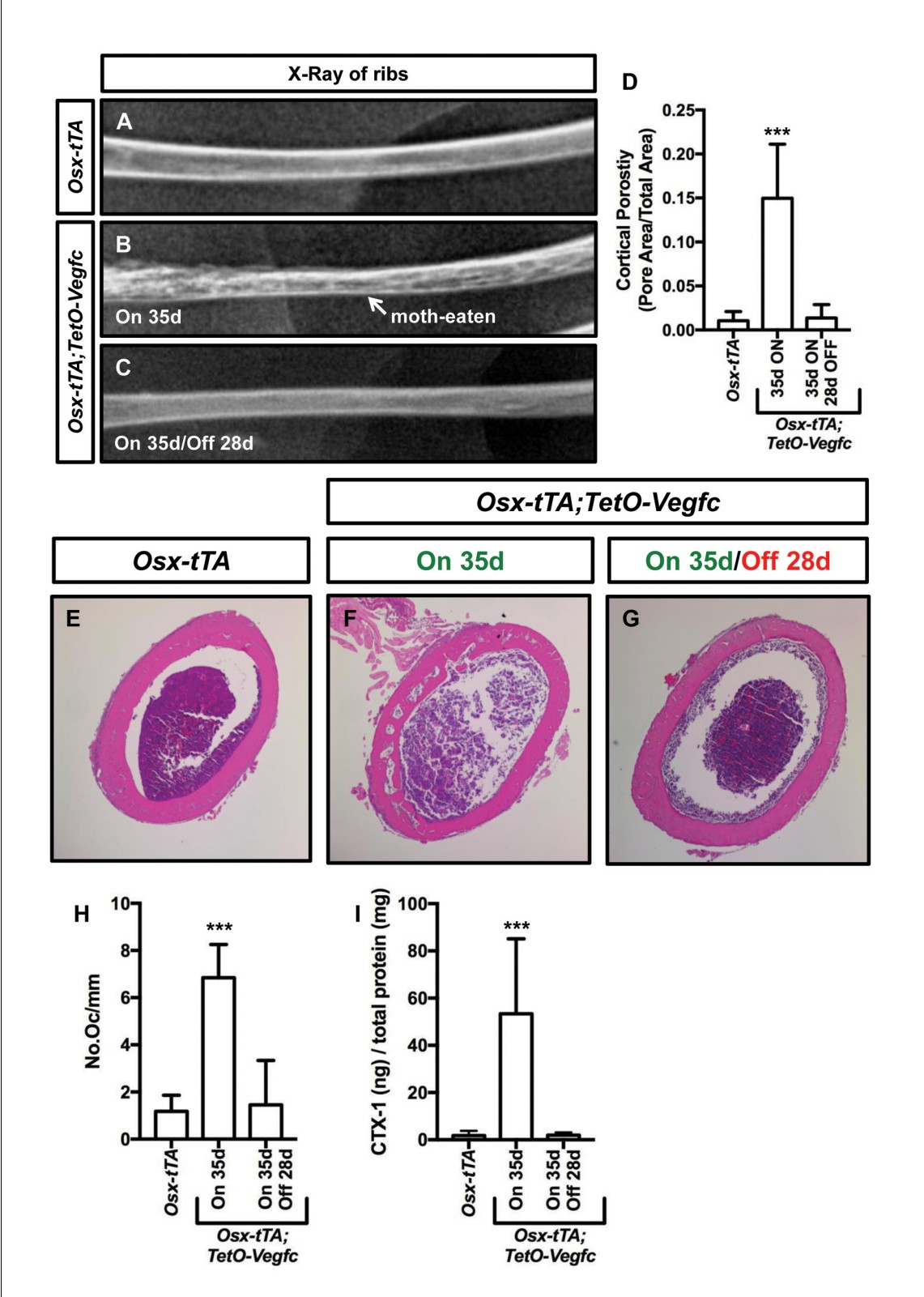

**Figure 7.** The bone phenotype of *Osx-tTA;TetO-Vegfc* mice is reversible. (**A–C**) Representative x-ray images of femurs from *Osx-tTA* mice, *Osx-tTA; TetO-Vegfc* mice that received normal water from E18.5 to P35 (On 35d), and *Osx-tTA;TetO-Vegfc* mice that received normal water from E18.5 to P35 and then doxycycline water from P35 to P63 (On 35d/Off 28d). *Osx-tTA;TetO-Vegfc* femurs switched from a moth-eaten appearance to a normal appearance following exposure to doxycycline. (**D**) Graph showing cortical bone porosity values for femurs from *Osx-tTA* mice (0.011 ± 0.0103; n = 4),
*Figure 7 continued on next page*

*Figure 7 continued*

*Osx-tTA;TetO-Vegfc* mice that received normal water from E18.5 to P35 (0.149 ± 0.0615; n = 7), and *Osx-tTA;TetO-Vegfc* mice that received normal water from E18.5 to P35 and then doxycycline water from P35 to P63 (0.014 ± 0.0152; n = 5). (E–G) Representative images of H and E stained femurs. The femur from the *Osx-tTA;TetO-Vegfc* mouse that received normal water from E18.5 to P35 is filled with pores. (H) Graph showing the number of osteoclasts per mm of bone for *Osx-tTA* mice (1.18. ±0.6818; n = 4), *Osx-tTA;TetO-Vegfc* mice that received normal water from E18.5 to P35 (6.84 ± 1.41; n = 4), and *Osx-tTA;TetO-Vegfc* mice that received normal water from E18.5 to P35 and then doxycycline water from P35 to P63 (1.45 ± 1.88; n = 4). (I) Graph showing CTX-1 values for *Osx-tTA* mice (1.7 ± 2.045; n = 8), *Osx-tTA;TetO-Vegfc* mice that received normal water from E18.5 to P35 (53.3 ± 31.8; n = 7), and *Osx-tTA;TetO-Vegfc* mice that received normal water from E18.5 to P35 and then doxycycline water from P35 to P63 (1.9 ± 1.19; n = 4). (***$p<0.001$, ANOVA followed by Dunnett's multiple comparisons test. Values were tested against values for *Osx-tTA* mice.).
DOI: https://doi.org/10.7554/eLife.34323.014

will help distinguish between these two possibilities and shed light on the molecular players and pathways that control the patterning of the lymphatic vasculature.

Transgenic manipulation of the VEGF-C/VEGFR3 signaling pathway has been widely used to investigate the role lymphatics serve in various pathological settings. Transgenic mice that express a soluble version of VEGFR3 have been used to study the pathophysiology of lymphedema (*Mäkinen et al., 2001*; *Rutkowski et al., 2010*), whereas VEGF-C transgenic mice have been used to model pulmonary lymphangiectasia (*Yao et al., 2014*), to study the pathophysiology of chylothorax (*Nitschké et al., 2017*), and to identify the specific function lymphatics serve in inflammation (*Huggenberger et al., 2011*) and cancer (*Mandriota et al., 2001*; *Hirakawa et al., 2007*). We have found that transgenic overexpression of VEGF-C in bone stimulates the formation of bone lymphatics and osteoclast-mediated bone resorption. Importantly, our transgenic mice can be used to further study these processes, which are relevant to the pathogenesis of GSD.

## Materials and methods

### Key resources table

| Reagent type (species) or resource | Designation | Source or reference | Identifiers | Additional information |
|---|---|---|---|---|
| Genetic reagent (*Mus musculus*) | *Osx-tTA* | PMID: 16854976 | | |
| Genetic reagent (*Mus musculus*) | *TetO-Vegfc* | PMID: 18988807 | | |
| Genetic reagent (*Mus musculus*) | *TetO-Vegfd* | PMID: 27342876 | | |
| Antibody | anti-GFP (chicken polyclonal) | abcam, ab13970 | | (1:1000) |
| Antibody | anti-Lyve-1 (goat polyclonal) | R and D Systems, AF2125 | | (1:250) |
| Antibody | anti-Podoplanin (hamster monoclonal) | abcam, ab11936 | | (1:1000) |
| Antibody | anti-VEGFR2 (rat monoclonal) | Eli Lily | DC101 | Function blocking antibody |
| Antibody | anti-VEGFR3 (rat monoclonal) | Eli Lily | mF4-31C1 | Function blocking antibody |
| Commerical assay, kit | ELISA kit - CTX-1 | CUSABIO, CSB - E12782M | | |
| Commerical assay, kit | ELISA kit - VEGF-C | CUSABIO, CSB - E07361M | | |
| Commerical assay, kit | TRIzol | Life Technologies, 15596018 | | |
| Commerical assay, kit | Rneasy RNA Isolation kit | Qiagen, 74104 | | |
| Commerical assay, kit | cDNA synthesis kit | BioRad, 74104 | | |

*Continued on next page*

*Continued*

| Reagent type (species) or resource | Designation | Source or reference | Identifiers | Additional information |
|---|---|---|---|---|
| Commerical assay, kit | 2 ml phase lock gel tube | Quanta, 2302830 | | |
| Commerical assay, kit | beads for tissue homogenizing | Biospec Products, 11079124zx | | |
| Chemical compound, drug | Doxycyline | Sigma Aldrich, D9891 | | |
| Chemical compound, drug | Zoledronic acid | SAGENT Pharmaceuticals, 801–66 | | |
| Software, algorithm | ImageJ | ImageJ 1.48 v | | |

## Mice and genotyping

The animal experiments described in this manuscript were carried out in accordance with an animal protocol approved by the Institutional Animal Care and Use Committee of UT Southwestern Medical Center. *TetO-Vegfc* mice (*Lohela et al., 2008*) were genotyped with the following primers: 5'-CCAAACCGGGCCCCTCTGCTAAC-3' and 5'-ACTGTCCCCTGTCCTGGTATTGAG-3'. The PCR product for the transgenic allele was approximately 450 bp. *TetO-Vegfd* mice (*Lammoglia et al., 2016*) were genotyped with the following primers: 5'-GCTCGTTTAGTGAACCGTCAG-3' and 5'-TGCTCGGATCTGTTGTTCAG-3'. The PCR product for the transgenic allele was approximately 250 bp. *Osx-tTA* mice (*Rodda and McMahon, 2006*) were genotyped with the following primers: 5'-CCTGGAAAATGCTTCTGTCCG-3', 5'-CAGGGTGTTATAAGCAATCCC-3', 5'-CAATGGTAGGCTCACTCTGGGAGATGAT −3', and 5'-AACACACACTGGCAGGACTGGCTAGG-3'. The wild-type band was 300 bp and the Cre band was 400 bp.

## Doxycycline administration

*Osx-tTA* (control) and *Osx-tTA;TetO-Vegfc* littermates received water containing doxycycline (1 mg/ml; Sigma-Aldrich, D9891, St. Louis, MO) and sucrose (5% w/v) from E0.5 to E18.5. Mice were then placed on normal water. For the reversal experiments, *Osx-tTA;TetO-Vegfc* mice were placed back on water containing doxycycline (1 mg/ml) and sucrose (5% w/v). Doxycycline water was replaced three times a week.

## Therapy studies with mice

DC101 and mF4-31C1 are rat monoclonal antibodies that block VEGFR2 and VEGFR3, respectively (ImClone/Eli Lilly). P21 *Osx-tTA;TetO-Vegfc* mice were treated with sterile saline (3x/week; i.p.), DC101 (800 µg; 3x/week; i.p.), or mF4-31C1 (800 µg; 3x/week; i.p.) for 2 weeks. To assess the effect of osteoclast inhibition on bone loss, P21 *Osx-tTA;TetO-Vegfc* mice were treated with sterile saline (q.a.d.; i.p.) or zoledronic acid (1.2 µg; q.a.d.; i.p.) for 2 weeks. Mice were randomly assigned to the various treatment groups.

## Immunohistochemistry

Slides were deparaffinized with xylene and rehydrated through a descending EtOH series. Endogenous peroxidase activity was blocked by incubating slides with hydrogen peroxide diluted in methanol. Antigen retrieval was performed by incubating slides in a proteinase K solution at 37°C for 5 min. Slides were washed with PBS and then blocked for 1 hr with TBST (TBS +0.2% Tween 20)+20% Aquablock. Tissue sections were incubated overnight with primary antibodies diluted in TBST +5% BSA. The following primary antibodies were used for immunohistochemistry: goat anti-Lyve-1 (R and D Systems; AF2125, Minneapolis, MN), chicken anti-GFP (Abcam; ab13970, Cambridge, MA), and hamster anti-podoplanin (Abcam; ab11936). Slides were washed with TBST and then incubated with HRP conjugated secondary antibodies diluted in TBST +5% BSA. Slides were washed with TBST and antibody binding was detected with DAB (Dako, K3468, Santa Clara, CA).

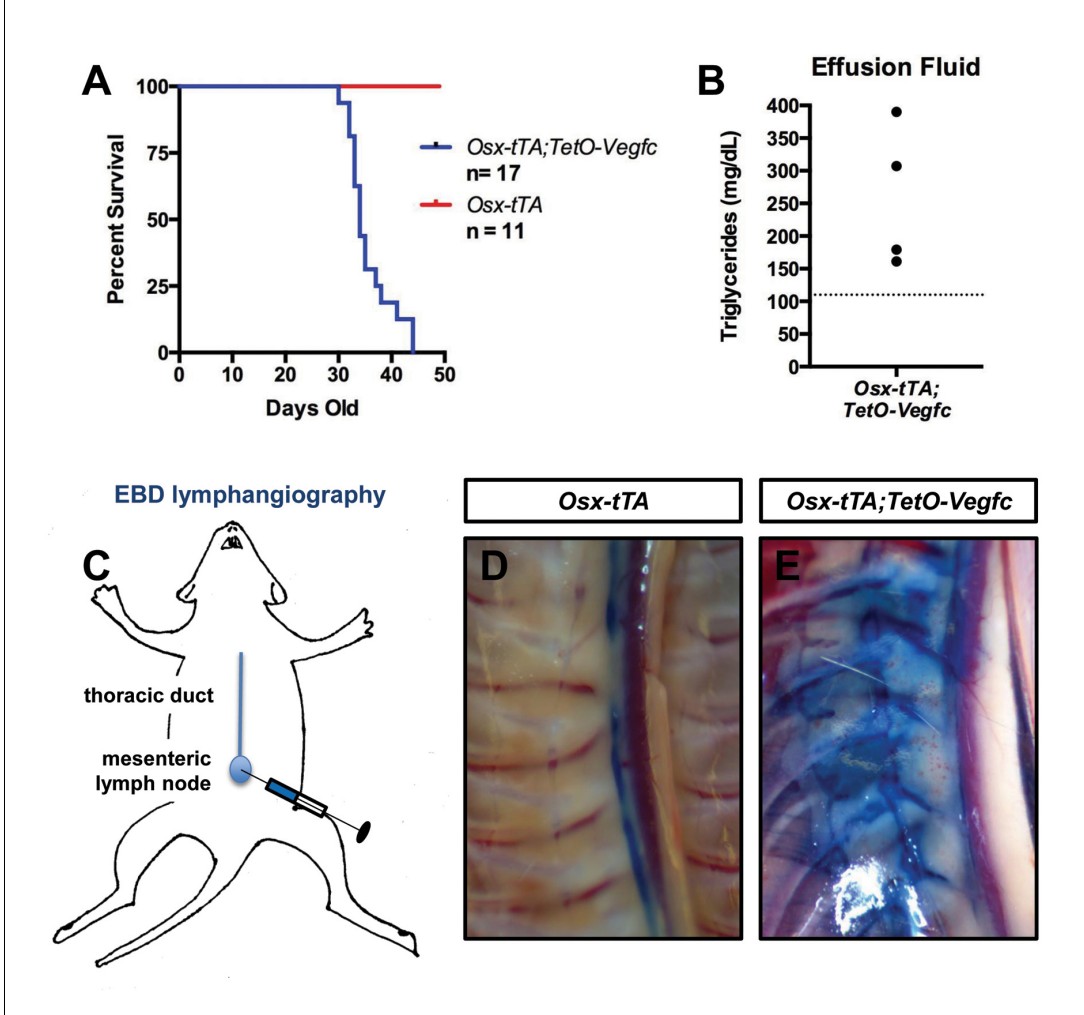

**Figure 8.** *Osx-tTA;TetO-Vegfc* mice develop chylothorax. (**A**) Survival curve for *Osx-tTA* and *Osx-tTA;TetO-Vegfc* mice (p<0.0001 Log-rank (Mantel-Cox) test). (**B**) Graph showing triglyceride levels in effusion fluid collected from *Osx-tTA;TetO-Vegfc* mice (n = 4). The dashed line marks 110 mg/dl. (**C**) Overview of the Evans blue dye (EBD) lymphangiography method. EBD injected into the mesenteric lymph node is transported to the thoracic duct. (**D**) EBD is confined to the thoracic duct in *Osx-tTA* mice (n = 4). (**E**) In *Osx-tTA;TetO-Vegfc* mice, EBD spills from the thoracic duct into periosseous lymphatics in muscle (n = 4).

DOI: https://doi.org/10.7554/eLife.34323.015

## Measuring lymphatic vessel index

To quantify bone lymphatics, multiple pictures of cortical bone and of the area below the growth plate were analyzed in ImageJ. A grid (19,000 cm$^2$) was placed over the pictures and the number of times that gridlines intersected within or on a lymphatic was determined.

## Tartrate-resistant acid phosphatase (TRAP) staining and analysis

Bone sections were deparaffinized with xylene, rehydrated through a descending EtOH series, and then placed in pre-warmed TRAP staining solution (TRAP basic incubation media + Fast Red Violet Salt + Napthol AS-MX Phosphate) for 1 hr. Slides were rinsed with water and counterstained with Fast Green for 30 s. Next, slides were rinsed with water, dehydrated through an ascending EtOH series, and cleared in xylene. Pictures were taken with an AmScope FMA050 camera attached to a Nikon Eclipse E600 microscope. The number of osteoclasts per mm bone surface was assessed with ImageJ.

## ELISA for VEGF-C and CTX-1

Plasma was collected from mice and commercially available ELISA kits were used to measure the circulating levels of VEGF-C (CUSABIO, #CSB-E07361M, Houston, TX) and CTX-1 (CUSABIO, #CSB-E12782M). A BCA assay was performed to determine the total protein content of each plasma sample. VEGF-C and CTX-1 values were normalized to the total protein content.

## RNA isolation and quantitative PCR

Frozen tibias were placed inside an RNase-free bag and crushed with a hammer. The crushed samples were transferred to a tube containing 1 ml of TRIzol (Life Technologies, 15596018, Carlsbad, CA) and 1 ml of beads (Biospec Products, #11079124zx, Bartlesville, OK). The samples were homogenized in a Minibead Beater (Biospec Products). The homogenized lysate was transferred to a 2 ml Phase Lock Gel Tube (Quanta, #2302830) and spun at 12,000 $g$ for 5 min. The upper layer of the spun solution was transferred to a Qiagen column and RNA was isolated with an RNeasy kit (Qiagen) according to the manufacturer's instructions. cDNA was synthesized with an iScript cDNA synthesis kit (BioRad; #74104) according to the manufacturer's instructions. The following primers were used in SYBR green qPCR reactions to amplify *Vegfc* (5'-TCCCCTGTCCTGGTATTGAG-3' and 5'-CGAGG TCAAGGCTTTTGAAG-3') and *beta-actin* (5'-CTGTCGAGTCGCGTCCA-3' and 5'-ACCCA TTCCCACCATCACAC-3'). Relative VEGF-C mRNA levels were calculated by the $2^{-\Delta\Delta CT}$ method.

## Evans blue dye lymphangiography

Our lymphangiography experiments were modeled after an approach recently published by Nitschke and colleagues (*Nitschké et al., 2017*). Immediately after euthanasia, the mesenteric lymph node was identified and injected with Evans blue dye (1% w/v). The thoracic cavity was then opened and pictures of the thoracic duct and chest wall were captured with an AmScope FMA050 camera.

## Assessment of cortical bone

Fixed bones were scanned using a µCT imaging system (35, Scanco Medical, Bassersdorf, Switzerland). The µCT Evaluation Program (V6.6) was used to measure the volume of the pores in bone and the total volume of the region of interest. Cortical porosity was determined by dividing the total volume of the pores by the total volume of the region of interest. Cortical bone was also assessed histologically. Decalcified bones were sectioned with a microtome and stained with hematoxylin and eosin. Images were captured with an AmScope FMA050 camera attached to a Nikon Eclipse E600 microscope. ImageJ was used to measure the area of the pores in bone and the total area of the region of interest. Cortical porosity was determined by dividing the total area of the pores by the total area of the region of interest.

## Three-point bending assay

Peak load at failure was tested in femora using a 3-point bending technique. Femora were placed with the anterior aspect up so that an actuator contacted the bones at mid-diaphysis. A Test Resources DDL200 axial loading machine outfitted with an Interface SMT1-22 force transducer was set to a cross-head displacement rate of 0.1 mm/sec. The femora rested on two supports 5 mm apart.

## Osteo-assay surface plate

One ml of primary human LECs (50,000 cells/ml) and one ml of RAW264.7 cells (50,000 cells/ml) were seeded into separate wells of an osteo-assay plate (Corning; #3987, Corning, NY). Recombinant RANKL (final concentration 50 ng/ml) was added to the wells that contained RAW264.7 cells. Cells were cultured at 37°C for 72 hr. Media was aspirated and one ml of 10% bleach was added to each well. Wells were then washed with water and images were captured with an inverted microscope. Image J was used to measure the area resorbed by cells.

## Statistical analysis

Data were analyzed using GraphPad Prism statistical analysis software (Version 7.0). All results are expressed as mean ± SD. The number of mice in each group is indicated in the figure legends (n = number of mice). For experiments with two groups, unpaired student's T-tests were performed to test means for significance. For experiments with more than two groups, differences were

assessed by ANOVA followed by Dunnett's multiple comparisons test. Data were considered significant at $p < 0.05$.

## Acknowledgements

We thank Jack Kelly (President, Lymphangiomatosis and Gorham's Disease Alliance) and Tiffany Ferry (President, The Lymphatic Malformation Institute) for their valuable comments on the project. We also thank Rolf Brekken and members of JMST for their comments on the manuscript. Funding: This work was supported by the Department of Surgery at UT Southwestern Medical Center (MTD) and by a grant from The Lymphatic Malformation Institute (MTD).

## Additional information

### Competing interests

Kari Alitalo: Reviewing editor, *eLife*. Bronislaw Pytowski: During preparation of the manuscript, B Pytowski was an employee of Eli Lilly and continues to hold stock in the company. The other authors declare that no competing interests exist.

### Funding

| Funder | Grant reference number | Author |
| --- | --- | --- |
| The Lymphatic Malformation Institute | Research Grant | Michael T Dellinger |
| The Department of Surgery at UT Southwestern Medical Center (MTD) | | Michael T Dellinger |

The funders had no role in study design, data collection and interpretation, or the decision to submit the work for publication.

### Author contributions

Devon Hominick, Investigation, Writing—review and editing; Asitha Silva, Noor Khurana, Ying Liu, Paul C Dechow, Jian Q Feng, Investigation; Bronislaw Pytowski, Joseph M Rutkowski, Resources; Kari Alitalo, Resources, Writing—review and editing; Michael T Dellinger, Conceptualization, Formal analysis, Supervision, Investigation, Methodology, Writing—original draft, Writing—review and editing

### Author ORCIDs

Michael T Dellinger http://orcid.org/0000-0002-3315-4239

### Ethics

Animal experimentation: The animal experiments described in this manuscript were carried out in accordance with animal protocols (2014-0031 and 2016-101510) approved by the Institutional Animal Care and Use Committee of UT Southwestern Medical Center.

### Decision letter and Author response

Decision letter https://doi.org/10.7554/eLife.34323.018
Author response https://doi.org/10.7554/eLife.34323.019

## Additional files

### Supplementary files

• Transparent reporting form
DOI: https://doi.org/10.7554/eLife.34323.016

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
