## [Decision Letter]

Thank you for submitting your article "VEGF-C promotes the development of lymphatics in bone and bone loss" for consideration by *eLife*. Your article has been favorably evaluated by Didier Stainier (Senior Editor) and four reviewers, one of whom, Holger Gerhardt (Reviewer #1), is a member of our Board of Reviewing Editors. The following individuals involved in review of your submission have also agreed to reveal their identity: Joyce Bischoff (Reviewer #2); Baptiste Coxam (Reviewer #4).

The reviewers have discussed the reviews with one another and the Reviewing Editor has drafted this decision to help you prepare a revised submission.

Summary:

The presented study shows that transgenic overexpression of VEGF-C in murine bone induces abnormal formation of lymphatic vessels in the bone, where they are not normally present. This is associated with increased number and activity of osteoclasts and bone loss. Intriguingly, the phenotype can be reversed after inhibition of VEGF-C expression. The *OsxtTA;TetO-Vegfc* mouse model generated in this study thus recapitulates features of, and provides the first animal model for a human disease called Gorham-Stout disease (GSD). The authors provide a comprehensive and convincing characterization of the phenotype, and the study raises many interesting questions, some of which are discussed by the authors.

Essential revisions:

The reviewers all find the work of substantial interest and great value for the GSD community as a model that may allow for deeper mechanistic understanding of the pathogenesis and therefore possible treatment of this disease. However, they all expressed concerns that in its present form, the manuscript is not strong enough in two points, a) the validity of the model for GSD, (i.e. is VEGF-C the driver of GSD in patients and are the lymphatics directly the cause of bone degradation, or is there a direct effect of VEGF-C on osteoclasts involved?) and b) the utility of the model to understand pathogenesis for example by investigating the origin of the ectopic lymphatic vessels. Following discussions, the reviewers agreed that the latter point may represent a future direction beyond the scope of a revision and only represents added value if the former point, the validity of the model, is sufficiently substantiated. They therefore request the following essential revisions:

1) Further validation as GSD model ideally on patient material if possible (including VEGF-C and osteoclast staining). The reviewers recognise however that obtaining patient material of good quality may be challenging.

2) Deeper investigation into the functional consequences of the performed VEGFR2 and 3 inhibition including assessment of osteoclasts, TRAP staining, and bone porosity as well as bone strength measurement. Given that reduction of osteoclasts using Zoledronic acid treatment (Figure 4) reduces bone destruction without affecting lymphatic vessels, it is critical to understand whether a treatment that primarily reduces lymphatics will affect osteoclasts and bone destruction. This should help to untangle possible direct from indirect effects of VEGF-C on bone.

---

## [Author Response]

Essential revisions:

The reviewers all find the work of substantial interest and great value for the GSD community as a model that may allow for deeper mechanistic understanding of the pathogenesis and therefore possible treatment of this disease. However, they all expressed concerns that in its present form, the manuscript is not strong enough in two points, a) the validity of the model for GSD, (i.e. is VEGF-C the driver of GSD in patients and are the lymphatics directly the cause of bone degradation, or is there a direct effect of VEGF-C on osteoclasts involved?) and b) the utility of the model to understand pathogenesis for example by investigating the origin of the ectopic lymphatic vessels. Following discussions, the reviewers agreed that the latter point may represent a future direction beyond the scope of a revision and only represents added value if the former point, the validity of the model, is sufficiently substantiated. They therefore request the following essential revisions:1) Further validation as GSD model ideally on patient material if possible (including VEGF-C and osteoclast staining). The reviewers recognise however that obtaining patient material of good quality may be challenging.

GSD patient samples are very rare and they are frequently decalcified with acid so a pathologist can analyze them. This makes it very difficult to perform molecular studies and specific stains (e.g. TRAP staining) with GSD patient samples. We have tried to obtain high-quality patient material for several years. Unfortunately, we do not have samples that would allow us to determine whether VEGF-C is locally elevated in patients with GSD. Although we haven’t been able to measure VEGF-C in patients with GSD, we believe that our mice will be of great value to the GSD research community. Our transgenic mice can be used to answer fundamental questions regarding the biology of bone lymphatics. We believe that this knowledge will significantly advance our understanding of processes that are relevant to the pathophysiology of GSD.

2) Deeper investigation into the functional consequences of the performed VEGFR2 and 3 inhibition including assessment of osteoclasts, TRAP staining, and bone porosity as well as bone strength measurement. Given that reduction of osteoclasts using Zoledronic acid treatment (Figure 4) reduces bone destruction without affecting lymphatic vessels, it is critical to understand whether a treatment that primarily reduces lymphatics will affect osteoclasts and bone destruction. This should help to untangle possible direct from indirect effects of VEGF-C on bone.

To address the comment, we measured osteoclast number and bone porosity in femurs from vehicle and mF4-31C1-treated mice. We focused our analysis on mF4-31C1-treated mice because these mice do not develop bone lymphatics. Importantly, we found that mF4-31C1 inhibited osteoclastogenesis and bone loss in *Osx-tTA;TetO-Vegfc* mice. We have added these new findings to the manuscript.

Results: “To determine whether inhibition of lymphangiogenesis could prevent osteoclastogenesis and bone loss in *Osx-tTA;TetO-Vegfc* mice, we analyzed femurs from vehicle and mF4-31C1-treated mice. […] These data show that inhibition of lymphangiogenesis can prevent osteoclastogenesis and bone loss in *Osx-tTA;TetO-Vegfc* mice.”

Figure Legend 5: “Figure 5. mF4-31C1 inhibits osteoclast formation and bone loss in *Osx-tTA;TetO-Vegfc* mice. […] (G) Graph showing cortical bone porosity of femurs for vehicle-treated (0.07244 ± 0.02468, n=5) and mF4-31C1-treated (0.006375 ± 0.007087, n=4) mice. (** *P* < 0.01, unpaired student’s T-test)”

We measured bone strength in *Osx-tTA* and *Osx-tTA;TetO-Vegfc* mice by performing a three-point bending assay. This assay was performed in the fall of 2017 and was the last experiment we performed before submitting the manuscript for review. Importantly, the three-point bending assay is performed with freshly isolated unfixed samples. We did not perform the three-point bending assay with samples collected from mice that received vehicle, DC101, or mF4-31C1 because that animal experiment was performed in 2015; before we knew there were differences in bone strength between *Osx-tTA* and *Osx-tTA;TetO-Vegfc* mice. Unfortunately, we do not have unfixed samples from vehicle, DC101, or mF4-31C1-treated mice for the three-point bending assay. However, we have been able to show that mF4-31C1-treated mice have less porous bones than vehicle-treated mice (Figure 5). Based on this observation, we predict that mF4-31C1-treated mice would have stronger bones than vehicle-treated mice.